# Diversity of Bacterial Soft Rot-Causing *Pectobacterium* Species Affecting Cabbage in Serbia

**DOI:** 10.3390/microorganisms11020335

**Published:** 2023-01-29

**Authors:** Aleksandra Jelušić, Petar Mitrović, Sanja Marković, Renata Iličić, Predrag Milovanović, Slaviša Stanković, Tatjana Popović Milovanović

**Affiliations:** 1Institute for Multidisciplinary Research, University of Belgrade, Kneza Višeslava 1, 11030 Belgrade, Serbia; 2Institute for Field and Vegetable Crops, National Institute of the Republic of Serbia, Maksima Gorkog 30, 21000 Novi Sad, Serbia; 3Faculty of Agriculture, University of Novi Sad, Trg Dositeja Obradovića 8, 21000 Novi Sad, Serbia; 4Agrosava doo, Palmira Toljatija 5, 11070 Belgrade, Serbia; 5Faculty of Biology, University of Belgrade, Studentski Trg 16, 11000 Belgrade, Serbia; 6Institute for Plant Protection and Environment, Teodora Drajzer 9, 11040 Belgrade, Serbia

**Keywords:** bacterial soft rot, cabbage, *Pectobacterium* spp., MLSA, rep-PCR, pathogenicity, virulence

## Abstract

The aim of this work was to identify and characterize the pectolytic bacteria responsible for the emergence of bacterial soft rot on two summer cabbage hybrids (Cheers F1 and Hippo F1) grown in the Futog locality (Bačka, Vojvodina), known for the five-century-long tradition of cabbage cultivation in Serbia. Symptoms manifesting as soft lesions on outer head leaves were observed during August 2021, while the inner tissues were macerated, featuring cream to black discoloration. As the affected tissue decomposed, it exuded a specific odor. Disease incidence ranged from 15% to 25%. A total of 67 isolates producing pits on crystal violet pectate (CVP) medium were characterized for their phenotypic and genotypic features. The pathogenicity was confirmed on cabbage heads. Findings yielded by the repetitive element palindromic-polymerase chain reaction (rep-PCR) technique confirmed interspecies diversity between cabbage isolates, as well as intraspecies genetic diversity within the *P. carotovorum* group of isolates. Based on multilocus sequence typing (MLST) using genes *dnaX*, *mdh*, *icdA*, and *proA*, five representative isolates were identified as *Pectobacterium carotovorum* (Cheers F1 and Hippo F1), while two were identified as *Pectobacterium versatile* (Hippo F1) and *Pectobacterium odoriferum* (Hippo F1), respectively, indicating the presence of diverse *Pectobacterium* species even in combined infection in the same field. Among the obtained isolates, *P. carotovorum* was the most prevalent species (62.69%), while *P. versatile* and *P. odoriferum* were less represented (contributing by 19.40% and 17.91%, respectively). Multilocus sequence analysis (MLSA) performed with concatenated sequences of four housekeeping genes (*proA*, *dnaX*, *icdA*, and *mdh*) and constructed a neighbor-joining phylogenetic tree enabled insight into the phylogenetic position of the Serbian cabbage *Pectobacterium* isolates. Bacterium *P. odoriferum* was found to be the most virulent species for cabbage, followed by *P. versatile*, while all three species had comparable virulence with respect to potato. The results obtained in this work provide a better understanding of the spreading routes and abundance of different *Pectobacterium* spp. in Serbia.

## 1. Introduction

Cabbage (*Brassica oleracea* var. *capitata* L.) is one of the world’s most important vegetable crops, as its high yield and adaptability have facilitated worldwide distribution [1]. Owing to the abundance of health-promoting phytochemicals (glucosinolates, polyphenols, vitamins, and proteins), cabbage features in the traditional cuisine of many countries and has also been traditionally used for medicinal purposes [2]. According to the data provided by the Food and Agriculture Organization Corporate Statistical Database (FAOSTAT) for 2020, globally, about 2,414,288 ha were designated for cabbage cultivation [3]. 

Futog, a suburban settlement located in the province of Vojvodina (Serbia), has been known for cabbage cultivation since the 18th century [4]. However, numerous pests and pathogens currently threaten cabbage production both in Serbia and worldwide, with *Pectobacterium* species (fam. Enterobacteriaceae), the causative agents of bacterial soft rot disease, being particularly harmful. The symptoms of infection with these pectinolytic enterobacteria may be found in the cabbage fields during cultivation, as well as post-harvest, during transport, storage, and marketing, leading to considerable yield reduction and economic losses [5]. According to Bhat [6], bacterial soft rot causes greater total post-harvest yield loss than any other bacterial disease. Losses on cabbage caused by soft rot alone, or in combination with other storage issues, are estimated at 25–50% in a single season in the USA, with the greatest loss occurring in New York and Wisconsin [7]. However, the extent of yield loss is influenced by pathogen−host specificity, as well as numerous external factors (temperature, humidity, etc.) [8]. Although bacterial soft rot is a disease that affects only agricultural and horticultural crops, the presence of the soft rot-causing Pectobacteriaceae is not only limited to host plants but extends to the weed species growing near the infected fields, as well as water, air, and soil, thus serving as a potent source of inoculum [9]. In addition, soft rot-causing bacteria can survive in infected fleshy organs during storage, as well as in debris, on roots or other parts of host plants, and in the pupae of several insects [10]. Thus, the most likely entry routes for the pathogen involve either roots or natural openings/mechanically created wounds on the upper plant parts. Once inside the plant, the soft rot-causing bacteria begin to move and multiply in the intercellular spaces, softening the affected tissues and transforming them into a slimy mass. The slimy mass extrudes through cracks in the epidermis into the soil, or to other plants that are placed in close proximity during storage, which subsequently become infected [10]. Bacterial survival in nature is dependent on various factors. For instance, their survival in soil depends on soil pH, temperature, moisture, magnesium concentration, and calcium nutrition and mostly takes up to six months in the absence of plant debris [11]. Temperature greatly affects the survival as well as the pathogenicity of *Pectobacterium* species [12].

Scientific interest in *Pectobacteria* primarily stems from their broad host range and an emerging genetic diversity associated with dynamic evolutionary processes, mainly occurring through gene acquisition, genome rearrangement, and gene loss [13]. Achtman [14] pointed out that higher rates of divergence in *Pectobacterium* spp. were detected on metabolic genes compared to the ribosomal operon and posited that the observed changes may be related to the adaptation of strains to specific environmental niches. Therefore, typing and analysis of several protein-coding loci (multilocus sequence typing and analysis, MLST/MLSA) could allow the degree of phenotypic relatedness to be established more reliably than is possible by analyzing 16S rRNA or some nonprotein-coding genes [15]. In extant research, different combinations of housekeeping genes have been used for MLST and MLSA. For example, Ma et al. [16] examined seven genes (*acnA*, *gapA*, *icdA*, *mdh*, *mtlD*, *pgi*, and *proA*) to reconstruct the phylogeny of the tested *Pectobacterium* and *Dickeya* strains, due to (i) their proven ability to provide sufficient sequence diversity for distinguishing closely related species, (ii) their ubiquity in most enterobacteria, and (iii) the involvement of their products in diverse aspects of bacterial metabolism. Further, Moleleki et al. [17] and Pitman et al. [18] reconstructed the phylogeny of *P. wasabiae* strains obtained from potato grown in South Africa and New Zealand based on the *gapA* and *mdh*, and *acnA* and *mdh* genes, respectively. In their work, Waleron et al. [19] used a combination of nine housekeeping genes (*gapA*, *gyrA*, *icdA*, *pgi*, *proA*, *recA*, *recN*, *rpoA*, and *rpoS*) for MLSA of *P. zantedeschiae* strains isolated from calla lily in central Poland and Serbia. On the other hand, Marković et al. [20] used a combination of genes *dnaX*, *proA,* and *mdh* to identify the causal agents (*P. versatile* and *P. carotovorum*) of potato blackleg disease in Serbia. The gene *dnaX* was shown to reliably distinguish between *Pectobacterium* and *Dickeya* isolates in several prior studies focusing on the potato blackleg causal agents in different countries [21,22,23]. Considering the aforementioned factors (high discriminatory power and reproducibility) and previous experience gained in working with plant pathogens within the Enterobacteriaceae family, four housekeeping genes (i.e., *dnaX*, *icdA*, *mdh*, and *proA*) were utilized in the present study for typing the pectolytic isolates obtained from cabbage.

Repetitive element palindromic PCR (rep-PCR) is widely recognized as a useful technique for profiling different *Pectobacterium* species. In their study, Norman et al. successfully used rep-PCR with primers for BOX-, ERIC-, and REP-PCR for strain-level differentiation of *P. carotovorum* populations isolated from nursery retention ponds and large hypereutrophic lakes [24]. In addition to the aforementioned rep-PCR techniques (BOX-, ERIC-, and REP-PCR), for the first time, Maisuria and Nerurkar [25] also used GTG_5_-PCR, due to its highest discriminatory power, to differentiate *P. carotovorum* strains from soil and diseased fruits and vegetables. According to Zoledowska et al. [26], REP-PCR is the most useful tool for grouping *P. parmentieri* potato strains and is superior to BOX- and ERIC-PCR. Thus, as BOX-, ERIC-, REP-, and GTG_5_-PCR are among the most commonly used rep-PCR methods for DNA profiling and are proven to be sufficiently discriminative to reveal subtle differences even among strains belonging to the same species, all four methods were adopted in the present study.

According to Song et al. [27], *P. carotovorum* is ranked among the most common causative agents of cabbage soft rot. Nonetheless, *P. aroidearum*, *P. brasiliense*, *P. odoriferum*, *P. polaris*, and *P. wasabiae* species have also been reported to result in cabbage soft rot in different countries [28,29,30,31,32,33]. In Serbia, different *Pectobacterium* species (i.e., *P. atrosepticum*, *P. brasiliense, P. carotovorum, P. punjabense, P. zantedeschiae,* and *P. versatile*) have been isolated from broccoli, calla lily, carrot, celery, parsley, potato, squash, and watermelon [19,20,23,34,35,36,37,38,39]. 

Given the expanding genetic diversity of *Pectobacterium* spp. and the lack of recent data on the agents causing soft rot on cabbage in Serbia, the goal of this study was to contribute to their identification and characterization based on molecular and pathogenic features.

## 2. Materials and Methods

### 2.1. Sample Collection and Pathogen Isolation

In August 2021, bacterial soft rot symptoms were noted on the summer cabbage hybrids Cheers F1 (Takii Seed, Field I (geographic coordinates 45.2507100, 19.7247660)) and Hippo F1 (Sakata Seed, Field II (geographic coordinates 45.2567200, 19.7275470)) grown in two fields located in Futog (Bačka, Vojvodina) of 0.5 ha and 1 ha size, respectively. Soft rot symptoms on outer leaves, along with deep maceration of inner head leaves accompanied by black discoloration, were found on the inspected plants (Figure 1). Due to decomposition, the affected tissue exuded a specific odor. Disease incidence in the visited fields was estimated at 15−25%.

Prior to isolation, collected samples were washed under running tap water and dried on filter paper at room temperature. Isolations were performed on crystal violet pectate (CVP) media [40] from the small leaf sections (2–3 mm) that encompassed the transition zones between healthy and diseased tissue. CVP plates were incubated at 26 °C. All bacterial colonies forming characteristic cavities on the medium were selected and transferred onto nutrient agar (NA) [41] to obtain pure cultures. Isolates were long-term stored at −80 °C in lysogeny broth (LB) [42] supplemented with 30% (v/v) of sterile glycerol.

### 2.2. Pathogenicity on Cabbage

The pathogenicity of 67 isolates obtained in this study was evaluated using cabbage heads (unknown cultivar). Before inoculation, cabbage heads were washed under running tap water, uniformly sprayed with 70% ethanol, and dried at room temperature. Isolates used for inoculation were grown in nutrient broth (NB, HiMedia Laboratories) at 26 °C for 48 h while shaking and were adjusted to approximately 1 × 10^8^ CFU mL^−1^. Inoculations were performed by puncturing holes in cabbage heads and filling them with bacterial suspensions (~200 µL). The assays were performed in two sets of three independent replicates. Inoculated cabbage heads were placed in plastic boxes which were kept under room temperature (25 ± 1 °C) and high humidity (90−100%) conditions. Sterile distilled water (SDW) was used as a negative control, while the *P. carotovorum* strain Pcc10, previously isolated from cabbage in Bosnia and Herzegovina [8], served as a positive control treatment.

Cabbage heads were visually observed daily in order to monitor the occurrence of soft rot symptoms and the disease progress until complete decay. Emergence of soft lesions around the holes 24 h after the inoculation of cabbage heads with the suspension of tested isolates was considered a pectolytic-positive reaction.

### 2.3. Genotyping Methods

#### 2.3.1. DNA Extraction

Genomic DNA from the 67 cabbage isolates was extracted according to the hexadecyltrimethylammonium bromide (CTAB) procedure described previously by Popović et al. [43].

#### 2.3.2. Preliminary Identification

All cabbage isolates were preliminarily identified using specific primers (F0145/E2477) designed based on the partial sequence of gene *pmrA* (response regulator) of *P. carotovorum* [44]. The sequences of the used primers are listed in Table 1. PCR amplifications were performed in a mixture (25 µL) consisting of Thermo Scientific DreamTaq PCR Master Mix (2×) (12.5 µL), nuclease-free water (Thermo Scientific^TM^, Waltham, MA, USA) (9.5 µL), 10 µM primers (forward/reverse) (1 µL each), and sample DNA (1 µL), according to the conditions proposed by Kettani-Halabi et al. [44]. Presence of a band in the searched position of 666 bp was checked on 1% agarose gel in relation to the positive control *P. carotovorum* strain Pcc10 and 200–10,000 bp SmartLadder MW-1700-10 (Eurogentec). 

#### 2.3.3. Repetitive Element Palindromic PCR (Rep-PCR)

Genetic diversity among the obtained 67 cabbage isolates was evaluated using the rep-PCR fingerprinting method with two oligonucleotide primer pairs [ERIC1R/ERIC2 (ERIC-PCR) and REP1R-I/ REP2-I (REP-PCR)] and two single oligonucleotide primers [BOXA1R (BOX-PCR) and GTG_5_ (GTG_5_-PCR)] corresponding to the interspersed repetitive sequence elements. Primer sequences of the used primers are listed in Table 1. The PCR mixture (25 µL) comprised 12.5 µL of Thermo Scientific DreamTaq PCR Master Mix (2×), 9.5 µL of nuclease-free water (Thermo Scientific^TM^, Waltham, MA, USA), 1 µL of each (forward/reverse) primer (10 µM) and 1 µL of sample DNA. PCR amplifications for BOX-, ERIC-, and REP-PCR were performed under the conditions proposed by Louws et al. [45], while the methodology described by Versalovic et al. [46] was adopted for GTG_5_-PCR. After amplification, PCR products (5 µL) were mixed with DNA Gel Loading Dye (6X) (Thermo Scientific^TM^, Waltham, MA, USA) (2 µL) and were visualized on 2% agarose gel (FastGene^®^) strained with Midori Green Advance (Nippon Genetics Europe, Düren, Germany). PCR products were electrophoretically separated for 2.5 h (at 90 V and 300 mA) before being checked under a UV transilluminator. The obtained patterns were compared using PyElph 1.4 program and were subsequently used for the construction of the unweighted pair group method with arithmetic mean (UPGMA) phylogenetic tree. To easily compare the patterns obtained with each of the four used rep-PCR primers and to select all isolates that were potentially genetically different, each tree cluster, representing one DNA fingerprinting pattern, was provided with a number (DNA fingerprinting group). One isolate representing each combination of obtained DNA fingerprinting groups was randomly selected for further characterization (MLST and MLSA, and virulence assessment).

#### 2.3.4. Multilocus Sequence Typing and Analysis (MLST/MLSA)

DNA of the seven selected representative cabbage isolates (Pc2321, Pc3821, Pc4821, Pc5421, Pv6321, Po7521, and Pc8321) was amplified with the primers produced based on the partial sequences of four housekeeping genes (*dnaX* (dnaX-F/dnaX-R), *icdA* (icdA400F/icdA977R), *mdh* (mdh2/mdh4)*,* and *proA* (proAF1/proAR1)), encoding DNA polymerase III subunit tau, isocitrate dehydrogenase, malate dehydrogenase, and gamma-glutamyl phosphate reductase, respectively. Primer sequences of the used primers are listed in Table 1. For all reactions, PCR mixtures were created as described in the previous subsection for rep-PCR. PCR amplifications were performed under the conditions described by Sławiak et al. [22] for the gene *dnaX*, while those reported by Ma et al. [16] were adopted for the genes *icdA* and *proA*, and the strategy employed by Moleleki et al. [17] was utilized for the gene *mdh*. Presence of bands in the searched positions was checked on 1% agarose gel in relation to 200–10,000 bp SmartLadder MW-1700-10 (Eurogentec). PCR products were purified using the Qiagen QIAquick PCR Purification Kit before being sent to the Eurofins Genomics’ DNA sequencing service (Germany) for sequencing. The obtained sequences were manually checked for quality and were compared with the strains deposited in the National Center for Biotechnology Information (NCBI) database using the nucleotide BLAST (BLASTn) tool. All newly identified sequences were deposited to the NCBI database to obtain the accession numbers.

The phylogenetic position of the seven representative isolates was determined in relation to 19 strains of three *Pectobacterium* species, i.e., *P. odoriferum* (CFBP 1878, BC S7, and JK2.1), *P. carotovorum* (ATCC 15713, 25.1, WPP14, BP201601.1, JR1.1, XP-13, and Pcc2520), and *P. versatile* (14A, 3-2, SCC1, F131, DSM 30169, MYP201603, SR1, SR12, and Pv1520) isolated from different hosts (cabbage, carrot, Chinese cabbage, chicory, coleslaw, cucumber, kimchi cabbage, potato, and radish) and countries (Belarus, China, Denmark, Finland, France, Germany, Russia, Serbia, South Korea, and the USA) and retrieved from the NCBI database (Table 2). A neighbor-joining (NJ) phylogenetic tree was constructed based on the concatenated sequences (1639 nt) of genes *dnaX* (416 nt), *icdA* (479 nt), *mdh* (240 nt), and *proA* (504 nt). Before the tree construction, the sequences of all genes were aligned to the same size using the biological sequence alignment editor BioEdit 7.2 program and the ClustalW Multiple alignment tool. The tree was constructed in MEGA7 Software and was rooted with the *Dickeya solani* strain RNS 05.1.2A (Accession No. CP104920).

### 2.4. Virulence Assessment

The virulence potential of the seven representative cabbage isolates (Pc2321, Pc3821, Pc4821, Pc5421, Pv6321, Po7521, and Pc8321) was assessed using cabbage heads (unknown cv.) and potato tubers (cv. Arizona). Prior to inoculation, cabbage heads and potato tubers were sterilized as previously described (Section 2.2), and approximately 100 surface punctures (around 5 mm in depth) were made using a needle to allow bacteria to penetrate the tissue. Bacterial suspensions were prepared as described for the pathogenicity test (Section 2.2) and SDW was used as a negative control. Inoculations were performed by spraying cabbage heads/potato tubers as uniformly as possible, making sure to cover the entire outer surface. The experiment was conducted in three independent replicates, with three cabbage heads (nine in total per isolate) and five potato tubers (15 in total per isolate) for each replicate. Two experiments—for disease progress rating five and seven days after inoculation (d.a.i.), respectively—were performed in parallel. For both experiments, the initial weights of cabbage heads and potato tubers were recorded. To obtain statistically comparable results, sums of the initial weights for each of the tested cabbage isolates were almost equal. The inoculated cabbage heads and potato tubers were placed in plastic boxes under high humidity conditions (90−100%). The experiment was performed during summer when the room temperature was 28 ± 2 °C.

Cabbage heads and potato tubers were weighed, whereby the mean values of the initial and the final (5 and 7 d.a.i.) weights were used to calculate the disease progress curve (AUDPC) according to the following equation:(1)$$\mathrm{AUDPC}=\overset{n-1}{\underset{i=1}{\sum }}\left(\frac{y_{i}+y_{i+1}}{2}\right)\left(t_{i+1}-t_{i}\right)$$
where *y_i_* = disease progress assessment at the *i*th observation, *t_i_* = time (days) at the *i*th observation, and *n* = total number of observations. 

All obtained values were statistically processed using Minitab 21 Statistical Software, whereby one-way analysis of variance (one-way ANOVA) was performed, and the resulting values were compared using Tukey’s honestly significant difference (HSD) test. Values having *p* < 0.05 were considered statistically significant.

## 3. Results

### 3.1. Isolation, Preliminary Identification, and Pathogenicity

In this study, 67 isolates were selected from cabbage hybrids Cheers F1 (34 isolates) and Hippo F1 (33 isolates), forming characteristic cavities on CVP medium and small, whitish, irregularly shaped colonies on NA (Table 3). All isolates produced bands at 666 bp after amplification with the *Pectobacterium*-specific primer pair F0145/E2477 (Table 3).

The pathogenicity of all cabbage isolates was confirmed on cabbage heads by visually identifying irregularly shaped soft lesions (approximately 3−5 cm in diameter) around the inoculation points (holes) 1 d.a.i. The diameter of decomposing tissue enlarged daily, and the affected area spread from the outer leaves to the inner tissues, while causing tissue discoloration from cream to black. At 5 d.a.i., cabbage heads were almost completely macerated and exuded a specific odor. 

### 3.2. Genetic Characterization

#### 3.2.1. Rep-PCR

The UPGMA trees showing genetic diversity among the 67 tested cabbage isolates—constructed based on the obtained BOX-, ERIC-, GTG_5_-, and REP-PCR banding patterns—are shown in Appendix A, which also features virtual gel images of rep-PCR patterns corresponding to each group of isolates. Based on the differences found, each tree cluster was assigned a different number: BOX (I–VII), ERIC (I–VI), GTG_5_ (I-VII), and REP (I–VI) (Table 3). The obtained results indicate that the tested isolates are genetically diverse. BOX- and GTG_5_-PCR generated seven (I-VII), while ERIC- and REP-PCR generated six distinct banding patterns (I-VI). Based on the BOX- and GTG_5_-PCR findings, isolates were divided into the same seven groups, namely **I:** Pc2021–Pc2821, **II:** Pc3121–Pc3921, **III:** Pc4221–Pc4921, **IV:** Pc5021–Pc5721, **V:** Pv1021–Pv1621 and Pv6121–Pv6621, **VI:** Po7021–Po7521 and Po9121–Po9621, and **VII:** Pc8021–Pc8721 (Table 3). However, ERIC- and REP-PCR did not separate the isolates into six identical groups. ERIC-PCR implied the homogeneity of isolates Pc3121–Pc3921 and Pc4221–Pc4921 by placing them in the same tree cluster (DNA fingerprinting group **II**), while REP-PCR indicated homogeneity of isolates Pc4221–Pc4921 and Pc5021–Pc5721 (DNA fingerprinting group **III**) (Table 3). The distribution of the remaining isolates within the UPGMA groups remained the same and coincided with that obtained for BOX- and GTG_5_-PCR. Based on the combined results obtained with all four primers, seven isolates (Pc2321, Pc3821, Pc4821, Pc5421, Pv6321, Po7521, and Pc8321), each representing one group on the UPGMA tree (i.e., one DNA banding pattern), were randomly selected for further characterization.

#### 3.2.2. MLST and MLSA

Based on the NCBI BLASTn analysis, five representative cabbage isolates (Pc2321, Pc3821, Pc4821, Pc5421, and Pc8321) were identified as *P. carotovorum* (representing the group of isolates Pc2021, Pc2121, Pc2221, Pc2321, Pc2421, Pc2521, Pc2621, Pc2721, Pc2821, Pc3121, Pc3221, Pc3321, Pc3421, Pc3521, Pc3621, Pc3721, Pc3821, Pc3921, Pc4221, Pc4321, Pc4421, Pc4521, Pc4621, Pc4721, Pc4821, Pc4921, Pc5021, Pc5121, Pc5221, Pc5321, Pc5421, Pc5521, Pc5621, Pc5721, Pc8021, Pc8121, Pc8221, Pc8321, Pc8421, Pc8521, Pc8621, and Pc8721), one representative cabbage isolate (Pv6321) was identified as *P. versatile* (representing the group of isolates Pv1021, Pv1121, Pv1221, Pv1321, Pv1421, Pv1521, Pv1621, Pv6121, Pv6221, Pv6321, Pv6421, Pv6521, and Pv6621), and one isolate Po7521 was identified as *P. odoriferum* (representing the group of isolates Po7021, Po7121, Po7221, Po7321, Po7421, Po7521, Po9121, Po9221, Po9321, Po9421, Po9521, and Po9621), with the percent identity in the 97.76–100%, 99.78–100%, and 99.57–100% range, respectively, depending on the sequenced gene (*dnaX*, *icdA*, *mdh*, and *proA*), as shown in Table 4. Accordingly, *P. carotovorum* was the most prevalent species (62.69%), while *P. versatile* and *P. odoriferum* were less represented (contributing by 19.40% and 17.91%, respectively). The sequences obtained for the seven representative cabbage isolates were deposited in the GenBank under the following accession numbers: OP729211-OP729217 (*dnaX*), OP729218-OP729224 (*icdA*), OP729225-OP729231 (*mdh*), and OP729232-OP729238 (*proA*).

The NJ phylogenetic tree generated based on the concatenated sequences of genes *dnaX, icdA*, *mdh*, and *proA* is presented in Figure 2. Based on these genes, the tested and comparative *P. carotovorum*, *P. versatile*, and *P. odoriferum* isolates/strains were separated into three clusters within the tree, each corresponding to one species. However, genetic heterogeneity (i.e., intraspecies genetic diversity) was observed within each species. Five of the seven tested cabbage *P. carotovorum* isolates examined in this study were separated into four groups (**I:** Pc2321 and Pc4821, **II:** Pc3821, **III:** Pc5421, and **IV:** Pc8321) within the cluster. The remaining tested *P. versatile* isolate Pv6321 was the most closely related to the comparative *P. versatile* strain DSM 30169 isolated from cabbage in Germany, while the *P. odoriferum* isolate Po7521 was most similar to the type *P. odoriferum* strain CFBP 1878 isolated from chicory in France. *D. solani* strain RNS 05.1.2A was placed on a monophyletic tree branch as an outgroup.

### 3.3. Virulence Assessment 

The developed disease symptoms observed seven days after the spray-inoculation of cabbage heads and potato tubers with suspensions of the tested isolates are presented in Figure 3.

The results of the AUDPC analysis of the tested isolates on cabbage heads and potato tubers are shown in Figure 4A,B, respectively.

The AUDPC values pertaining to cabbage heads ranged from 4964.2 to 5990.46 for the *P. odoriferum* isolate Po7521 and *P. carotovorum* isolate Pc8321, respectively (Figure 4A). Based on the obtained AUDPC values, the *P. odoriferum* isolate Po7521 exhibited the highest virulence potential, followed by the *P. versatile* isolate Pv6321, while the *P. carotovorum* isolates Pc2321, Pc3821, Pc4821, Pc5421, and Pc8321 exhibited the lowest (and comparable) virulence potential. Statistically significant differences between the initial (0 d.a.i.) and the final (5 and 7 d.a.i.) weights were observed after cabbage inoculation with the *P. carotovorum* isolates Pc2321 and Pc3821, as well as the *P. odoriferum* isolate Po7521. On the other hand, the weights measured 5 and 7 d.a.i. with the *P. versatile* isolate Pv6321 were comparable but differed significantly from the initial weight (0 d.a.i.).

The AUDPC values pertaining to potato tubers ranged from 249.92 to 342.15 for the *P. carotovorum* isolates Pc2321 and Pc4821, respectively (Figure 4B). As the AUDPC values obtained for the seven representative isolates were comparable, all tested isolates appeared to be equally virulent with respect to this host. However, the weights measured 5 and 7 d.a.i. were statistically significantly lower than the initial weights when samples were inoculated with the *P. carotovorum* isolates Pc4821, Pc5421, and Pc8321. For the remaining two *P. carotovorum* isolates (Pc2321 and Pc3821), one *P. versatile* Pv6321, and one *P. odoriferum* Po7521, the weights measured 5 and 7 d.a.i. were comparable, but differed significantly from the initial values (0 d.a.i.).

## 4. Discussion

To the best of our knowledge, this is the first study since the pioneering work of Arsenijević and Obradović published more than 20 years ago [47], in which *Erwinia carotovora* subsp. *carotovora* was identified in the Bačka region, to provide evidence on the presence and diversity of the three bacteria (i.e., *P. carotovorum*, *P. odoriferum*, and *P. versatile*) causing soft rot in cabbage in Vojvodina (Serbia). The obtained results indicated presence of a combined infection in Field II, where all three identified species were confirmed on the cabbage hybrid Hippo F1. However, in Field I (hybrid Cheers F1), only *P. carotovorum* was detected. Analysis of the 67 cabbage isolates indicated that *P. carotovorum* was the most represented (62.69%), followed by *P. versatile* (19.40%) and *P. odoriferum* (17.91%). These findings are not surprising, given that *P. carotovorum* species is generally recognized as the main causal agent of soft rot in Brassicaceae plants [28]. This species was described on cabbage and Chinese cabbage in China, Brazil, Malaysia, Korea, and Bosnia and Herzegovina [8,48,49,50,51]. However, there is paucity of data on the presence of *P. versatile* and *P. odoriferum* on cabbage. In addition to *P. carotovorum*, *P. versatile*, and *P. odoriferum*, Chen et al. [31] also reported species *P. aroidearum*, *P. brasiliense*, and *P. polaris* on Chinese cabbage grown in different districts of Beijing (China). Presence of *P. odoriferum* was also reported on cabbage and Chinese cabbage in Central Poland, China, and Iran [28,48,52]. However, the lack of data on the presence of *P. odoriferum* and *P. versatile* on cabbage does not imply its absence on this host, as it is at least partly due to their recent reclassification (i.e., elevation from subspecies to the species level) within the genus *Pectobacterium* [53].

It is known that the genus *Pectobacterium* includes heterogonous strains characterized by diverse biochemical, physiological, and genetic properties even within the same species [54]. Based on the rep-PCR results, the *Pectobacterium* spp. isolates tested as a part of the present study were genetically heterogeneous, forming seven (BOX- and GTG_5_-PCR) and six (ERIC- and REP-PCR) groups on the UPGMA tree, depending on the utilized primers. In other words, the rep-PCR results indicate interspecies genetic heterogeneity, as well as intraspecies heterogeneity within *P. carotovorum* isolates only, which were clustered in five (BOX- and GTG_5_-PCR) and four (ERIC- and REP-PCR) groups. The rep-PCR (BOX-, ERIC-, and REP-PCR) analysis conducted by Alvarado et al. [49] also revealed high genetic variability among *P. carotovorum* strains isolated from Chinese cabbage in Pernambuco state (north-east Brazil). Based on the rep-PCR with primers for BOX-, ERIC-, and REP-PCR, Loc et al. [38] reached a similar conclusion to the one put forward in this study, positing that *Pectobacterium* strains of the same species tend to group closely according to their respective taxonomic designations. Considering that authors of several extant studies singled out rep-PCR as discriminative enough to reveal subtle differences between different *Pectobacterium* spp., as well as those among the same species, as proposed by Zoledowska et al. [26] for *P. parmentieri* strains from Poland, rep-PCR is a promising technique for the clarification of genetic diversity and discrimination of soft rot-causing *Pectobacterium* spp.

In the present study, typing of four housekeeping genes (*dnaX*, *icdA*, *mdh*, and *proA*) enabled appropriate identification of the tested cabbage isolates and a clear separation of each of the three identified species from one another. The existence of inter- and intra-species genetic heterogeneity between the three detected *Pectobacterium* species was again confirmed based on MLSA with concatenated sequences of the same four genes. According to Zeigler [55], among genes present in all sequenced bacterial genomes, the *dnaX* gene is considered one of the best candidates for assigning bacterial strains to the species level. Sławiak et al. [22] also highlighted the usefulness of gene *dnaX* for the identification of European potato *Dickeya* spp. strains. However, despite the demonstrated high resolution of the *dnaX* gene, the use of other protein-coding genes should not be discouraged due to the well-known claim that mutations, as the main engine of evolution, in *Pectobacterium* spp. mostly occur on these genes [14]. Moreover, using multiple genes allows for a greater genome coverage, which undoubtedly leads to a more reliable phylogeny reconstruction. In extant research, the remaining three genes (*icdA*, *mdh*, and *proA*) were successfully used for the typing of *Pectobacterium* spp. in different combinations with other housekeeping genes (e.g., *acnA*, *gapA*, *mtlD*, *pgi*, *recA*, *rpoS*, etc.) [16,17,18,19,20]. Important parameters when selecting these genes are (i) ubiquity in most enterobacteria, (ii) high discriminatory ability, and (iii) indispensable role in key metabolic processes [16].

In the virulence assessment assay performed as a part of this work, the lowest AUDPC values pertaining to cabbage heads (i.e., the highest virulence potential) was observed for the *P. odoriferum* isolate Po7521, followed by the *P. versatile* isolate Pv6321. On the other hand, the *P. carotovorum* isolates Pc2321, Pc3821, Pc4821, Pc5421, and Pc8321 were the least virulent, with a comparable virulence potential for cabbage. However, such statistical differences in aggressiveness among isolates/species were not detected on potato tubers. While these findings may be indicative of host−pathogen specificity in the case of bacteria *P. odoriferum* and *P. versatile*, more confident claims about such interactions would require more extensive studies performed on a larger number of known hosts. Bearing in mind the wide distribution, ubiquity, and polyphagous nature of *P. carotovorum* [56], and the resulting genetic diversity that enabled this species to survive and adapt to different ecological niches (i.e., hosts), it is likely that differences in aggressiveness will be observed between different hosts. In the study conducted by Li et al. [57], *P. odoriferum* strains isolated from Chinese cabbage exhibited much higher virulence potential for Chinese cabbage compared to the tested *P. carotovorum* and *P. brasiliensis* strains (measured 24 h and 30 h post-inoculation), all obtained from the same host. These authors did not observe statistically significant differences in the virulence potential between *P. carotovorum* and *P. brasiliensis* strains. However, it remains to be established whether the differences between species would be sustained for the duration of the disease progression. Moreover, based on the results reported by Waleron et al. [58], no statistically significant differences in the virulence potential on potato were observed between *P. carotovorum* and *P. odoriferum,* or between *P. carotovorum* and *P. brasiliensis*, based on the analyses performed 3 d.a.i. 

This research significantly contributes to the current knowledge of the diversity of pectolytic bacteria affecting cabbage in Serbia, which has so far remained unexplored despite their great importance.

## Figures and Tables

**Figure 1 microorganisms-11-00335-f001:**
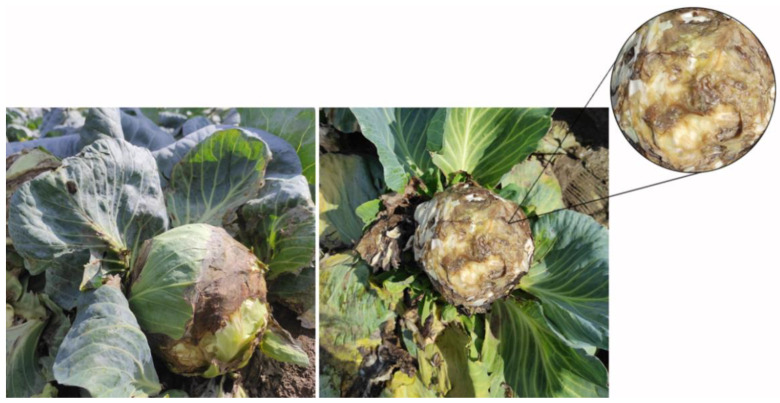
Bacterial soft rot symptoms in visited cabbage fields in Futog (Bačka, Vojvodina).

**Figure 2 microorganisms-11-00335-f002:**
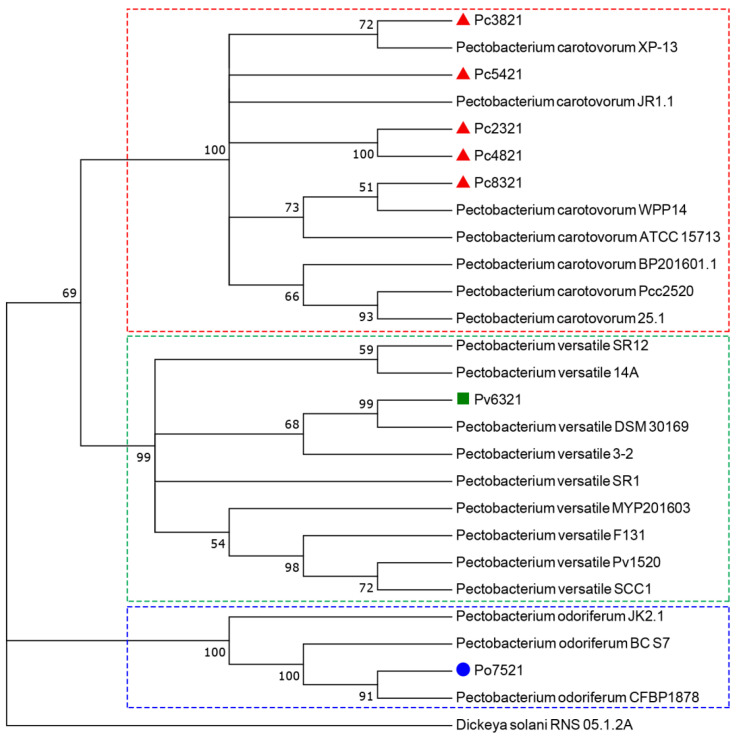
The neighbor-joining phylogenetic tree constructed based on the concatenated sequences of genes *dnaX*, *icdA*, *mdh*, and *proA* for seven representative *P. carotovorum*

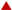
, *P. odoriferum*


, and *P. versatile*

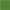
 isolates examined in this study and 19 strains of *P. carotovorum*, *P. odoriferum*, and *P. versatile* isolated from various hosts and countries, which were retrieved from the GenBank. The tree was rooted with the *D. solani* strain RNS 05.1.2A.

**Figure 3 microorganisms-11-00335-f003:**
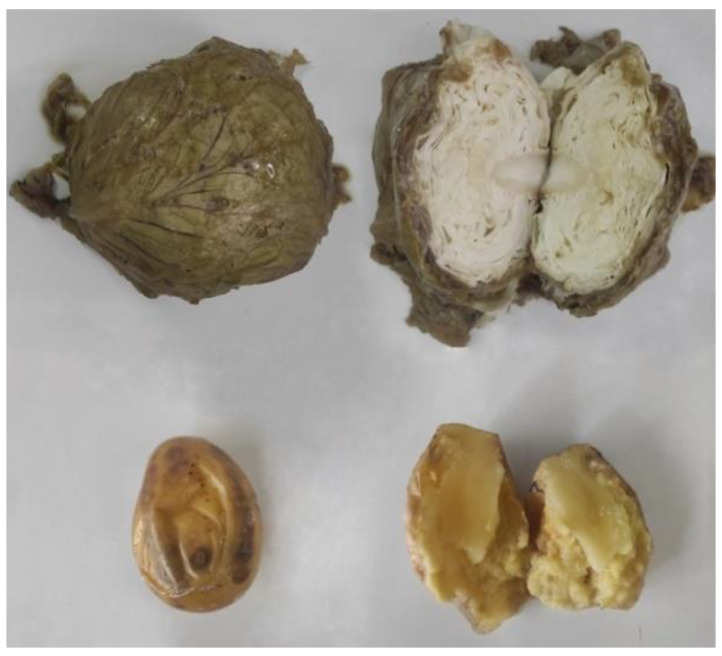
Examples of the bacterial soft rot symptoms on the cabbage head and potato tuber that were observed seven days after inoculation with the Serbian *P. versatile* strain Pv6321 and the *P. carotovorum* strain Pc3821, respectively.

**Figure 4 microorganisms-11-00335-f004:**
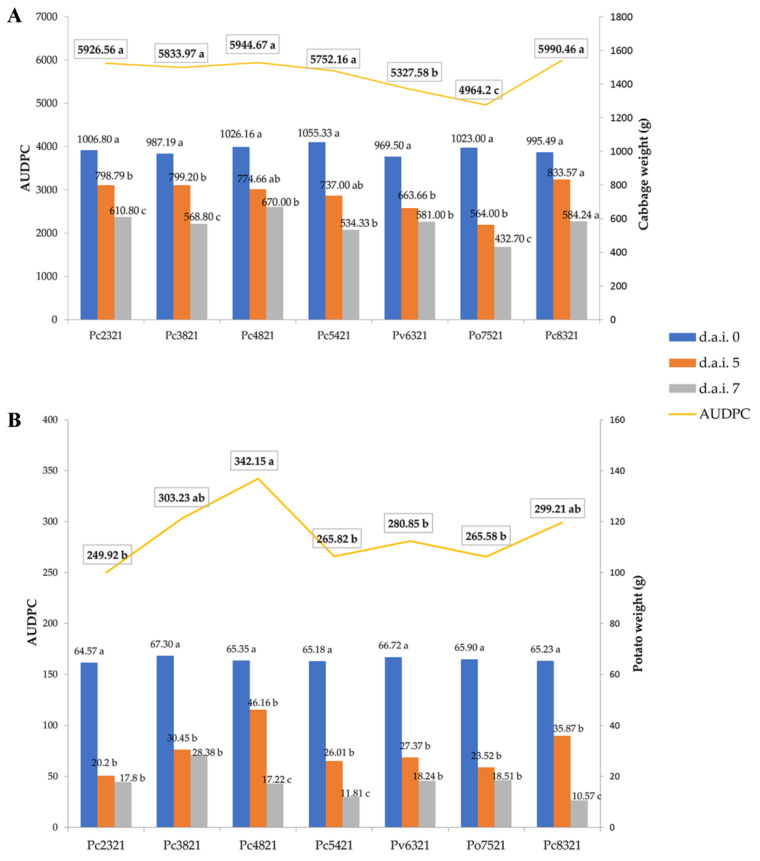
Disease progress curves (AUDPC) showing (**A**) cabbage and (**B**) potato weight ratings (obtained 5 and 7 d.a.i.) resulting from tissue maceration due to inoculation with seven representative *Pectobacterium* cabbage isolates. Different letters represent statistically significant differences.

**Table 1 microorganisms-11-00335-t001:** Primers used in this study.

Primer	Sequence (5′-3′)	Reference
***Pectobacterium*-specific primers**		
F0145	TACCCTGCAGATGAAATTATTGATTGTTGAAGAC	[44]
E2477	TACCAAGCTTTGGTTGTTCCCCTTTGGTCA	
**Primers for rep-PCR**		
ERIC1R	ATGTAAGCTCCTGGGGATTCAC	[45]
ERIC2	AAGTAAGTGACTGGGGTGAGCG	
REP1R-I	IIIICGICGICATCIGGC	
REP2-I	ICGICTTATCIGGCCTAC	
BOXA1R	CTACGGCAAGGCGACGCTGACG	
GTG_5_	GTGGTGGTGGTGGTG	[46]
**Primers for MLST**		
dnaX-F	TATCAGGTYCTTGCCCGTAAGTGG	[22]
dnaX-R	TCGACATCCARCGCYTTGAGATG	
icdA400F	GGTGGTATCCGTTCTCTGAACG	[16]
icdA977R	TAGTCGCCGTTCAGGTTCATACA	
proAF1	CGGYAATGCGGTGATTCTGCG	
proAR1	GGGTACTGACCGCCACTTC	
mdh2	GCGCGTAAGCCGGGTATGGA	[17]
mdh4	CGCGGCAGCCTGGCCCATAG	

**Table 2 microorganisms-11-00335-t002:** List of the comparative *Pectobacterium* spp. strains used for phylogenetic analysis with GenBank accession numbers.

Strain ^a^	Isolation Source	Locality	Year	GenBank Accession Numbers			
				*dnaX*	*icdA*	*mdh*	*proA*
** *Pectobacterium odoriferum* **							
CFBP 1878 ^T^	Chicory	France	1978	MK516907	JF926783	JF926793	JF926823
BC S7	Chinese cabbage	Beijing (China)	2007	CP009678	CP009678	CP009678	CP009678
JK2.1	Kimchi cabbage	South Korea	2016	CP034938	CP034938	CP034938	CP034938
** *Pectobacterium carotovorum* **							
ATCC 15713 ^T^	Potato	Denmark	1952	MW979047	FJ895850	FJ895851	FJ895853
25.1	Cucumber	Svietlahorsk (Belarus)	2009	CP088019	CP088019	CP088019	CP088019
WPP14	Potato	Wisconsin (USA)	2015	CP051652	CP051652	CP051652	CP051652
BP201601.1	Potato	Boseong (South Korea)	2016	CP034236	CP034236	CP034236	CP034236
JR1.1	Radish	South Korea	2016	CP034237	CP034237	CP034237	CP034237
XP-13	Potato	Zhouning (China)	2018	CP063242	CP063242	CP063242	CP063242
Pcc2520	Potato	Serbia	2020	MW805307	OP751390	OP751392	OP751393
** *Pectobacterium versatile* **							
14A	Potato	Minsk (Belarus)	1978	CP034276	CP034276	CP034276	CP034276
3-2	Potato	Minsk (Belarus)	1979	CP024842	CP024842	CP024842	CP024842
SCC1	Potato	Finland	1982	CP021894	CP021894	CP021894	CP021894
F131	Potato	Moscow (Russia)	1993	CP065030	CP065030	CP065030	CP065030
DSM 30169	Cabbage	Germany	2010	CP065143	CP065143	CP065143	CP065143
MYP201603	Potato	Miryang (South Korea)	2013	CP051628	CP051628	CP051628	CP051628
SR1	Carrot	Iowa, Ames (USA)	2019	CP084656	CP084656	CP084656	CP084656
SR12	Coleslaw	Iowa, Ames (USA)	2019	CP084654	CP084654	CP084654	CP084654
Pv1520	Potato	Serbia	2020	MW805306	OP751391	MZ682621	MZ682624

**^a^** Type strains are marked with the superscript T.

**Table 3 microorganisms-11-00335-t003:** Serbian cabbage *Pectobacterium* spp. isolates used in the present study, the hybrid they originate from, pectolytic activity, confirmation with *Pectobacterium*-specific primers, and DNA fingerprinting group affiliation.

Isolate Code	Hybrid	Pectolytic Activity	Genus Confirmation	DNA Fingerprinting Group			
				BOX	ERIC	GTG_5_	REP
Pc2021	Cheers F1	**+**	+	I	I	I	I
Pc2121	Cheers F1	+	+	I	I	I	I
Pc2221	Cheers F1	+	+	I	I	I	I
**Pc2321**	Cheers F1	+	+	I	I	I	I
Pc2421	Cheers F1	+	+	I	I	I	I
Pc2521	Cheers F1	+	+	I	I	I	I
Pc2621	Cheers F1	+	+	I	I	I	I
Pc2721	Cheers F1	+	+	I	I	I	I
Pc2821	Cheers F1	+	+	I	I	I	I
Pc3121	Cheers F1	+	+	II	II	II	II
Pc3221	Cheers F1	+	+	II	II	II	II
Pc3321	Cheers F1	+	+	II	II	II	II
Pc3421	Cheers F1	+	+	II	II	II	II
Pc3521	Cheers F1	+	+	II	II	II	II
Pc3621	Cheers F1	+	+	II	II	II	II
Pc3721	Cheers F1	+	+	II	II	II	II
**Pc3821**	Cheers F1	+	+	II	II	II	II
Pc3921	Cheers F1	+	+	II	II	II	II
Pc4221	Cheers F1	+	+	III	II	III	III
Pc4321	Cheers F1	+	+	III	II	III	III
Pc4421	Cheers F1	+	+	III	II	III	III
Pc4521	Cheers F1	+	+	III	II	III	III
Pc4621	Cheers F1	+	+	III	II	III	III
Pc4721	Cheers F1	+	+	III	II	III	III
**Pc4821**	Cheers F1	+	+	III	II	III	III
Pc4921	Cheers F1	+	+	III	II	III	III
Pc5021	Cheers F1	+	+	IV	III	IV	III
Pc5121	Cheers F1	+	+	IV	III	IV	III
Pc5221	Cheers F1	+	+	IV	III	IV	III
Pc5321	Cheers F1	+	+	IV	III	IV	III
**Pc5421**	Cheers F1	+	+	IV	III	IV	III
Pc5521	Cheers F1	+	+	IV	III	IV	III
Pc5621	Cheers F1	+	+	IV	III	IV	III
Pc5721	Cheers F1	+	+	IV	III	IV	III
Pv1021	Hippo F1	+	+	V	IV	V	IV
Pv1121	Hippo F1	+	+	V	IV	V	IV
Pv1221	Hippo F1	+	+	V	IV	V	IV
Pv1321	Hippo F1	+	+	V	IV	V	IV
Pv1421	Hippo F1	+	+	V	IV	V	IV
Pv1521	Hippo F1	+	+	V	IV	V	IV
Pv1621	Hippo F1	+	+	V	IV	V	IV
Pv6121	Hippo F1	+	+	V	IV	V	IV
Pv6221	Hippo F1	+	+	V	IV	V	IV
**Pv6321**	Hippo F1	+	+	V	IV	V	IV
Pv6421	Hippo F1	+	+	V	IV	V	IV
Pv6521	Hippo F1	+	+	V	IV	V	IV
Pv6621	Hippo F1	+	+	V	IV	V	IV
Po7021	Hippo F1	+	+	VI	V	VI	V
Po7121	Hippo F1	+	+	VI	V	VI	V
Po7221	Hippo F1	+	+	VI	V	VI	V
Po7321	Hippo F1	+	+	VI	V	VI	V
Po7421	Hippo F1	+	+	VI	V	VI	V
**Po7521**	Hippo F1	+	+	VI	V	VI	V
Po9121	Hippo F1	+	+	VI	V	VI	V
Po9221	Hippo F1	+	+	VI	V	VI	V
Po9321	Hippo F1	+	+	VI	V	VI	V
Po9421	Hippo F1	+	+	VI	V	VI	V
Po9521	Hippo F1	+	+	VI	V	VI	V
Po9621	Hippo F1	+	+	VI	V	VI	V
Pc8021	Hippo F1	+	+	VII	VI	VII	VI
Pc8121	Hippo F1	+	+	VII	VI	VII	VI
Pc8221	Hippo F1	+	+	VII	VI	VII	VI
**Pc8321**	Hippo F1	+	+	VII	VI	VII	VI
Pc8421	Hippo F1	+	+	VII	VI	VII	VI
Pc8521	Hippo F1	+	+	VII	VI	VII	VI
Pc8621	Hippo F1	+	+	VII	VI	VII	VI
Pc8721	Hippo F1	+	+	VII	VI	VII	VI

All isolates marked in bold are used as representatives in this study.

**Table 4 microorganisms-11-00335-t004:** The list of the 67 Serbian cabbage *Pectobacterium* spp. isolates and species affiliation, with the percent identity of the sequenced isolates based on the partial sequences of four sequenced genes (*dnaX*, *icdA*, *mdh*, and *proA*).

Isolate Code		Hybrid	Identification According to the NCBI BLASTn (Per. Ident)				
			Species	*dnaX*	*icdA*	*mdh*	*proA*
**Pc2321** (group Pc2021–Pc2821)		Cheers F1	*P. carotovorum*	99.79%	99.81%	99.75%	100%
**Pc3821** (group Pc3121–Pc3921)		Cheers F1	*P. carotovorum*	99.79%	100%	100%	99.25%
**Pc4821** (group Pc4221–Pc4921)		Cheers F1	*P. carotovorum*	100%	100%	99.75%	99.85%
**Pc5421** (group Pc5021–Pc5721)		Cheers F1	*P. carotovorum*	99.36%	99.44%	99.02%	97.76%
**Pv6321**	(group Pv1021–Pv1621, Pv6121–Pv6621)	Hippo F1	*P. versatile*	99.78%	100%	100%	99.85%
**Po7521**	(group Po7021–Po7521, Po9121–Po9621)	Hippo F1	*P. odoriferum*	99.79%	100%	100%	99.57%
**Pc8321** (group Pc8021–Pc8721)		Hippo F1	*P. carotovorum*	100%	100%	100%	97.91%

## Data Availability

Not applicable.

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
