# Peer review of "Diversity of Bacterial Soft Rot-Causing *Pectobacterium* Species Affecting Cabbage in Serbia"

_microorganisms, 2023, doi:10.3390/microorganisms11020335_

Round 1
Reviewer 1 Report
The paper of Jelusic et al., describes a comprehensive study performed with many Pectobacterium species that cause severe economic losses to cabbage cultivation in Serbia and world-wide.
To this aim, the Authors applied both molecular and pathogenicity assays to find out differences and similarities between the strains collection they used.
The paper is well written and the data obtained support the conclusions raised up by the Authors.
To better frame the significance of the study, I would suggest to provide also some information (in the Introduction or in the Discussion) about the epidemiological context that characterize Pectobacterium spp. (i.e., survival in soil and in plant, cycle of disease, and so on). This would allow the reader to have a more complete framework of the "soft rot" diseases caused by Pectobacterium species worldwide.
Minor comments:
Take care of the italics for Latin names. For example, at page 11: P. versatile; at page 12: P: odoriferum.
Reviewer 2 Report
The manuscript titled “Diversity of bacterial soft rot causing Pectobacterium species “ is devoted to identification of pectolytic bacteria responsible for the emergence of bacterial soft rot on two summer cabbage hybrids (Cheers F1 and Hippo F1) grown in Futog locality (Bačka, Vojvodina). A total of 67 isolates making pits on Crystal Violet Pectate (CVP) medium were characterized for their phenotypic and genotypic features. The pathogenicity was confirmed on cabbage heads. The repetitive element palindromic-polymerase chain reaction (rep-PCR) technique revealed interspecies diversity between cabbage isolates, as well as intraspecies genetic diversity within P. carotovorum group of isolates. Multilocus sequence typing (MLST) using genes dnaX, mdh, icdA, and proA identified five representative isolates as Pectobacterium carotovorum (Cheers F1 and Hippo F1), one as Pectobacterium versatile (Hippo F1), and one as Pectobacterium odoriferum (Hippo F1), indicating the presence of diverse Pectobacterium species, even in combined infection in the same field. Among the obtained isolates P. carotovorum was the most prevalent species (62.69%), while P. versatile and P. odoriferum were less represented (19.40 and 17.91%, respectively). Results obtained in this work provide a better understanding of the spreading routes and abundance of different Pectobacterium spp. in the Serbian field.
Despite the large work done to get the aims of this research, the manuscript needs significant improvements:
1) English language must be corrected by a native English speaker. Many sentences have structure inherited from original language of the text.
2) Some information in introduction is unrelated to the tasks of this research and can be removed (see other comments below).
3) Part “Material and methods” must be re-arranged.
4) Despite the good CVP medium selective work, identity of all isolates must be confirmed by Pectobacterium-specific PCR primers or by biochemical/serological analysis. PCR fingerprinting do not generate Pectobacterium-specific patterns, and some isolates besides MLST-identified ones could be of other pectolytic bacteria species.
5) Data presented in the Figure 2 must be transformed into Table with PCR fingerprint -based grouping (for each primer) for each particular isolate. It is important to include into PCR fingerprinting type strains of the target Pectobacterium species as reference.
6) Trees of MLST analysis must be re-built in traditional for publication form (rectangular) with cut-off nodes with bootstrap below 50%.
7) The three constructed by TCS haplotype networks to show a correlation between geographic origin and the determined haplotype cannot be treated as reliable because only a few strains for each remote geographic region were used and this part must be re-analyzed using much more strains (at least 10 for each analyzed region). Generally, this is not within the aim of this research as it stated from the title, and can be done in the next publication.
Some minor notes:
Line 42
Cabbage (Brassica oleracea var. capitata L.) is among the world’s most important vegetable crops, having worldwide distribution due to its pronounced adaptability [1].
Please, suggest such change: Cabbage (Brassica oleracea var. capitata L.) is one of the world’s most important vegetable crops, having worldwide distribution due to its high yield and adaptability [1].
Line 49
Thereof, harvested area of cabbage in Serbia was estimated at 7547 ha (Vojvodina: 1071 ha) and 7513 ha (Vojvodina: 1057 ha) in 2020 and 2021, respectively, with corresponding yields of 23.8 t/ha (Vojvodina: 29.3 t/ha) and 24.7 t/ha (Vojvodina: 30.1 t/ha) [4].
- This information can be removed without significant loss for the manuscript.
Line 53
Apart from being known for growing a specific cabbage cultivar called “Futoški“, fertile land distinguishes this area as crucial for agricultural production [5].
- - This information can be removed without significant loss for the manuscript.
Line 65
In view of the economic losses linked with soft rot disease, species P. carotovorum and P. atrosepticum are classified among the top 10 scientifically and economically important plant pathogenic bacteria [9].
- The reference 9 (Top 10 plant pathogenic bacteria in molecular plant pathology 2012) represents “the ranked list of bacteria as voted for by plant bacteriologists associated with the journal Molecular Plant Pathology.” Thus, it is important to show actual yield loss caused by the pathogens based on the real statistical data (especially for Srbija and close regions.
Line 73
Therefore, typing and analysis of several protein-coding loci (Multilocus sequence typing and analysis, MLST/MLSA) could enable measuring of a degree of phenotypic relatedness more reliably than analyses of 16S rRNA or some nonprotein-coding genes [12].
-Please, describe more MSLT/MLSA analysis for Pectobacterium spp. with detailed examples and justify your choice of the genes for sequencing. The same for PCR-fingerprinting.
Line 76.
According to Gardan et al. [13], P. carotovorum is ranked among the most common causals of cabbage soft rot.
The reference 13 (Gardan, et al. Elevation of three subspecies of Pectobacterium carotovorum to species level: Pectobacterium atrosepticum sp. nov., Pectobacterium betavasculorum sp. nov. and Pectobacterium wasabiae sp. nov. Int J Syst Evol 530 2003) has no original information about actual frequency of Pectobacterium species on cabbage. Please, find more relevant source.
Line 88
In August 2021, bacterial soft rot symptoms were noticed on summer cabbage hybrids Cheers F1 (Takii Seed, field I) and Hippo F1 (Sakata Seed, field II) grown in two fields located in Futog (Bačka, Vojvodina), on an area of 0.5 ha and 1 ha in size, respectively.
-Please, show actual geographic coordinates of this site.
Line 91
Symptoms were in the form of soft lesions on outer head leaves and maceration of inner tissues having cream to black discoloration (Figure 1).
- Suggested change: Soft rot symptoms on outer leaves and deep maceration of inner head leaves with black discoloration were found on the inspected plants (Figure 1).
Line 96
Isolations were performed on Crystal Violet Pectate (CVP) media from the small leaf sections (2–3 mm) that encompassed transition zones between healthy and diseased tissue. CVP plates were incubated at 26 °C. All bacterial colonies forming characteristic cavities on the medium were selected and transferred onto Nutrient Agar (NA) to obtain pure cultures. Isolates were long-term stored at ˗80 °C in Luria Bertani (LB) broth supplemented with 30% (v/v) of sterile glycerol.
- Please, give references for CVP, NA and LB. LB is for Lysogeny Broth according to Bertani (1951). Bertani, G. 1951. J. Bacteriol. 62:293. https://doi.org/10.1128/JB.62.3.293-300.1951
Line 106
Genomic DNA from the cabbage isolates was extracted according to the hexadecyltrimethylammonium bromide (CTAB) procedure described in the previous work of Popović et al. [29].
- If the CTAB method was used without changes there is no need to describe it at Line 108:
“For DNA extraction, isolates were grown on an NA medium for 48 h at 26 °C. Likewise obtained single colonies of each isolate were re-suspended in sterile distilled water and centrifugated for 5 min at 13,000 rpm. The obtained pellet was re-suspended in a mixture of TE buffer [10 mM Tris(hydroxymethyl) aminomethane (TRIS) and 1 mM Ethylenediaminetetraacetic acid (EDTA)], 10% Sodium Dodecyl Sulphate (SDS), and 20 mg/mL proteinase K and incubated at 37 °C for 30 min. Thereafter, DNA was purified with 5 M NaCl and 3% CTAB and the obtained mixture was heated at 65 °C for 20 min. DNA was extracted by adding chloroform. The upper phase, obtained after centrifugation (13,000 rpm for 10 min) was transferred to the clean 1.5 mL tubes. DNA was precipitated with 3M sodium acetate (pH 5.0) and ice-cold isopropanol. The mix was centrifuged for 15 min at 13,000 rpm. The final step consisted of washing DNA in 96% ice-cold ethanol and centrifugation for 10 min at 13,000 rpm. “
This text can be removed.
Line 122
Inter- and intra-species genetic diversity among the obtained 67 cabbage isolates was evaluated using the rep-PCR fingerprinting method with four primer pairs, BOXA1R (BOX-PCR), ERIC1R/ERIC2 (ERIC-PCR), GTG5 (GTG5-PCR), and REP1R-I/ REP2-I (REP-PCR)
- There are only 2 primer pairs and 2 single primers. Please, correct. So as you assay the Inter- and intra-species genetic diversity, reference strains of the species are essential.
Line 133
PCR products were electrophoretically separated for 2.5 h (90 V, 300 mA) and checked under a UV transilluminator.
- Please, describe the gel documentation and pattern recognition methods.
Line 141
DNA of the seven selected, representative cabbage isolates (Pc2321, Pc3821, Pc4821, Pc5421, Pv6321, Po7521, and Pc8321) was amplified with primers made based on the partial sequences of four housekeeping genes dnaX (dnaX-F/dnaX-R), icdA 1(icdA400F/icdA977R), mdh (mdh2/mdh4), and proA (proAF1/proAR1), encoding DNA polymerase III subunit tau, isocitrate dehydrogenase, malate dehydrogenase, and gamma-glutamyl phosphate reductase, respectively.
- Please, give references for the chosen MLST genes and justify your choice. Why these genes were selected among many used for enterobacteria? You can do it in the Introduction part.
Line 193
Table 2. List of the comparative Pectobacterium spp. strains used for phylogenetic and phylogeographic analysis with GenBank accession numbers.
- Please, move this table to the Supplement. See the main Note 7 (The three constructed by TCS haplotype networks to show a correlation between geographic origin and the determined haplotype cannot be treated as reliable because only a few strains for each remote geographic region were used and this part must be re-analyzed using much more strains (at least 10 for each analyzed region). Generally, this is not within the aim of this research as it stated from the title, and can be done in the next publication.) If you intend to keep this part, justify the choice of GenBank accessions for comparison.
Line 196
2.3. Pathogenicity and virulence 2.3.1. Pathogenicity on cabbage And Line 214
2.3.2. Virulence assessment
- this parts must be moved to the beginning of Materials and methods (after bacteria isolation). It is more correct to analyze genetic variation AFTER pathogenicity and virulence analysis.
Line 233
Mean values of initial and final (five and seven dai)
-Please, explain meaning of "dai" after the first use (is it d.a.i.? – days after inoculation)
Line 275
Figure 2. Unweighted pair group method with arithmetic mean (UPGMA) phylogenetic trees of tested cabbage Pectobacterium spp. isolates and virtual gel images representing rep-PCR fingerprinting patterns for each of the obtained groups of isolates for (A) BOX-PCR, (B) ERIC-PCR, (C) GTG5-PCR, and (D) REP-PCR.
- It is difficult to analyze this figure. Transform all data into one table with every isolate origin, PCR-fingerprinting groups and pathogenicity&virulence data. Add the genus confirmation test data for each used isolate (PCR, biochemical or serological test).
Line 298
Table 4. GenBank accession numbers of the seven representative Pectobacterium spp. isolates from this study.
- This table cam be transformed into text. There is no need to type each accession number – just write as OP729211- OP729215.
Line 320
Figure 3. The neighbor-joining phylogenetic tree made based on the concatenated sequences of genes icdA, mdh, and proA for seven representative P. carotovorum , P. odoriferum , and P. versatile isolates from this study and 32 strains of P. carotovorum, P. odoriferum, P. versatile, P. polaris, and P. brasiliense isolated from various hosts and countries, retrieved from the GenBank. The tree was rooted with D. solani strain RNS 05.1.2A.
- It is not clear why P. polaris, and P. brasiliense were used for analysis but other Pectobacterium species were ignored? Please, explain or remove them (P. polaris, and P. brasiliense).
Line 395
Figure 7. Disease progress curves (AUDPC) showing cabbage and potato weight loss ratings resulted by macerating tissue due to inoculation with 7 representative Pectobacterium cabbage isolates (scored at 5 and 7 dai). Different letters represent significant statistical differences.
- Please, use different Y axis scales for AUDPC and potato weight loss ratings.
Round 2
Reviewer 2 Report
The manuscript was significantly improved and can be published in this form.